# Carved Turn Control with Gate Vision Recognition of a Humanoid Robot for Giant Slalom Skiing on Ski Slopes

**DOI:** 10.3390/s22030816

**Published:** 2022-01-21

**Authors:** Cheonyu Park, Baekseok Kim, Yitaek Kim, Younseal Eum, Hyunjong Song, Dongkuk Yoon, Jeongin Moon, Jeakweon Han

**Affiliations:** 1Department of Convergence Robot System, Hanyang University, 55 Hanyangdaehak-ro, Sangnok-gu, Ansan-si 15588, Gyeonggi-do, Korea; cheonyu1@naver.com (C.P.); kbs0112000@gmail.com (B.K.); kimpd1224@naver.com (Y.K.); shealeum@gmail.com (Y.E.); 2Department of Mechanical and Aerospace Engineering, New York University, New York, NY 10003, USA; hs3927@nyu.edu; 3ERICA IUCF, Hanyang University, 55 Hanyangdaehak-ro, Sangnok-gu, Ansan-si 15588, Gyeonggi-do, Korea; solpi@wizgene.com; 4Sports Engineering Laboratory, Department of Physical Education, Seoul National University, 1 Gwanak-ro 38-gil, Gwanak-gu, Seoul 08732, Korea; moon00@snu.ac.kr; 5Department of Robotics, Hanyang University, 55 Hanyangdaehak-ro, Sangnok-gu, Ansan-si 15588, Gyeonggi-do, Korea

**Keywords:** ski robot, humanoid, carved turn, ZMP control, vision recognition

## Abstract

The performance of humanoid robots is improving, owing in part to their participation in robot games such as the DARPA Robotics Challenge. Along with the 2018 Winter Olympics in Pyeongchang, a Skiing Robot Competition was held in which humanoid robots participated autonomously in a giant slalom alpine skiing competition. The robots were required to transit through many red or blue gates on the ski slope to reach the finish line. The course was relatively short at 100 m long and had an intermediate-level rating. A 1.23 m tall humanoid ski robot, ‘DIANA’, was developed for this skiing competition. As a humanoid robot that mimics humans, the goal was to descend the slope as fast as possible, so the robot was developed to perform a carved turn motion. The carved turn was difficult to balance compared to other turn methods. Therefore, ZMP control, which could secure the posture stability of the biped robot, was applied. Since skiing takes place outdoors, it was necessary to ensure recognition of the flags in various weather conditions. This was ensured using deep learning-based vision recognition. Thus, the performance of the humanoid robot DIANA was established using the carved turn in an experiment on an actual ski slope. The ultimate vision for humanoid robots is for them to naturally blend into human society and provide necessary services to people. Previously, there was no way for a full-sized humanoid robot to move on a snowy mountain. In this study, a humanoid robot that transcends this limitation was realized.

## 1. Introduction

For robots to be useful in society, they must be able to perform all the activities that humans can perform [1]. If this is not possible owing to limitations associated with the form of robots, their widespread application is restricted. Therefore, the development of humanoids that resemble human shapes and can best imitate human behavior will greatly contribute to the enhanced utility of robots. However, since humanoids employ low-stability movement called bipedal walking, more research is needed to improve their stability. Moreover, the usefulness of humanoid robots will increase only when their ability to mimic human behavior under harsh environmental conditions is established via rigorous experimentation.

Skiing is a human behavior that is typically performed under extremely harsh environmental conditions, such as cold temperatures and snow. If a humanoid can successfully perform this activity under harsh conditions, this is a further upgrade of the robot’s capabilities. If robots can be used in such a difficult working environment, they can serve as very useful tools in rescues or for the exploration of unfamiliar areas [2].

Currently, research on special types of robots that can perform tasks on behalf of human counterparts in various environments and circumstances, such as exploration [3,4], crop collecting [5,6], and disasters [7,8,9], is being actively conducted. However, these special types of robots have limited universal applicability because of their specialized form. Therefore, research on the use of humanoids under different circumstances, such as walking [10,11,12,13] or the climbing of stairs and ladders [14,15,16], is ongoing. The objective is to facilitate the performance of various universal tasks by imitating human behavior. The Defense Advanced Research Projects Agency (DARPA) held the DARPA Robotics Challenge (DRC) from 2012 to 2015 to address this aim by creating various situations that could occur at nuclear power plants. At DRC, various types of robots, including humanoids, were tested on challenging tasks such as driving vehicles, getting out of cars, opening doors, turning valves, drilling walls, breaking through rough terrain, and climbing stairs. Various robots, such as ATLAS, developed by Boston Dynamics (Team WPI-CMU [17] and Team IHMC [18]), Valkyrie (NASA JSC [19]), DRC-HUBO + (Team KAIST [20], Team DRC-hubo@UNLV [21]), and THORMANG, developed by Robotis(Team ROBOTIS, Team SNU [22]), have been developed to successfully complete the aforementioned tasks.

In addition, research based on humanoid robots is actively being conducted in sports fields such as soccer and basketball. In particular, the annual Robo Cup is a world-class event in which robots developed by different teams around the world compete in a variety of sporting events. Among the various leagues of RoboCup, the humanoid league began in 2002 and aims to improve robot performance via competition in robot soccer games. Ultimately, in 2050, the robot teams aim to compete in the human World Cup championship and win [23,24].

Skiing is also a challenging task for robots owing to limitations in the development of sensors [25,26]. In the field of sports science, systematic research on the mechanics of this activity has been conducted [27,28,29]. Research has been conducted on stability analysis via physical feedback data acquisition during skiing, using sensors mounted on the human body [30] and skiing algorithms [31]. Since researchers could obtain information on the force and position movement generated during this activity, research has been conducted to investigate skiing robots. In 2009, Lahajnar et al. developed a robot that could ski on actual slopes. However, this robot did not have an anthropogenic form. It consisted of only the lower body, with only four joints and one degree of freedom in the knee joints, without the upper body [32].

In 2017, the University of Manitoba studied the balancing and turning of skiing robots using a 45 cm tall, small humanoid robot. The study revealed that these robots could readily turn around actual snow slopes using simple controllers that control only the angle of the torso without ZMP calculation [33]. In 2019, Ajou University conducted a Webot simulation of ski balancing and turning with a 45 cm tall, small humanoid robot. Balancing was attempted using the ZMP calculation, and a virtual LiDAR sensor was used to recognize and turn the gates to perform a simulation that passed between the flags [34]. In 2018, Takuma Saga of Osaka University developed a robot with a simple structure to teach children skiing motion and posture [35].

Until now, skiing robots could not imitate actual adult-sized human behavior, such as developing only the lower body or using small humanoids. In skiing competitions, the participants pass through gates installed on the slopes. For robots to perform a similar action, artificial intelligence algorithms that recognize the flags and generate their paths must be developed. However, to the best of our knowledge, the development of skiing robots using AI has not been pursued to date. Therefore, in this study, we developed the world’s first full-sized humanoid skiing robot with a height of 1.23 m. It can recognize installed flags in an actual outdoor slope using deep learning and turn through the gates with a carved turn using the ZMP method generally used in bipedal robots. Using this technology, a test was conducted based on an Alpine ski giant slalom competition using the human Olympic ski rules on an actual slope of 100 m. Figure 1 shows an image of DIANA, the humanoid robot developed in this study.

## 2. Hardware

### 2.1. Mechanical Design

Skiing robots must have sufficient degrees of freedom (DOFs) to perform the sharp turns that are characteristic of skiing. Figure 2 represents the overall design of DIANA. DIANA was designed to have a total of 23 DOFs, as shown in Table 1, and to be 1.23 m in height and 30 kg in weight. The head region, where a stereo camera was mounted, was designed such that the camera can recognize flags more accurately while undergoing various changes in field of view during skiing with Roll-Pitch-Yaw 3 DOFs. The arm was designed to move the ski pole with X-Y-Z 3 DOFs. In particular, the action of skiing causes a change in the center of mass owing to the movement of the upper body, so DIANA was designed to move the upper body in the angulation (the lateral movements of creating angles at the waist) and rotation directions by enabling the structure of the waist to move in the roll and yaw directions. In addition, the lower body facilitated Roll-Pitch-Yaw-X-Y-Z 6 DOFs so that the ski plate could be located in the desired location and direction.

Figure 3 shows the specifications of the sensor that was used for the skiing robot DIANA, and the design of the sensor mounting unit. A light detection and ranging (LiDAR) sensor and stereo camera were mounted on the head so that the flags could be recognized and the distance to the flags could be measured. An inertial measurement unit (IMU) sensor was installed at the waist, as close to the center of mass as possible, so that the speed of the robot and the feedback during skiing could be measured. The shape of the pedal component was designed in accordance with the ISO 5355:2005 standard to be directly mounted on the ski binding, and F/T sensors were mounted on the feet to estimate the ZMP of the robot.

### 2.2. Water Proof Design

In Figure 4, the appearance of DIANA wearing a skiing suit manufactured for waterproofing is shown.

Skiing robots can encounter snow and rain on actual slopes. In addition, a waterproof design is essential for falling on a slope. A dedicated ski suit using a waterproof fabric was designed for the robot system. This suit was cut to the appropriate size and shape to minimize interference with the robot’s movement, and important components such as electrical parts were double-wrapped to improve waterproofing. In addition, zippers were used on both the front and rear sides of the suit to detach and attach from the robot easily. The suit was designed to cover the entire robot. As a result, no safety accidents occurred due to snow and water penetration during the two-month test period.

### 2.3. Electrical Design

Figure 5 shows a diagram of the electronic components of the humanoid robot, DIANA. DIANA requires a lightweight board that can run artificial intelligence models with low power because of its limited space, load capacity, and power. The electric unit consists of a component for object inference and a component for posture control. Given that numerous computations are required for object inference, two types of boards were used to facilitate CPU resource management and to prevent accidents caused by delays in motor control operations. Both boards operate on the same network via the LAN hub.

NVIDIA’s Jetson TX2 is an embedded module that can operate at 7.5 W and is suitable for inference using CUDA processors. The environmental package for image inference was the JetPack 3.3 package provided by NVIDIA, in line with the specifications of Jetson TX2. By attaching Stereo Labs’ Zed Stereo Camera to Jetson TX2, distance can be measured using the phase difference between the two cameras, as in humans. Intel’s Joule was the board in charge of posture control and it was implemented by connecting the IMU, LiDAR, F/T sensor, and Usb2Dynamixel. The computer used for machine learning to create a learning model was equipped with three NVIDIA Titan xp, CUDA 9.2.148, CUDNN 7.2, TensorRT 4.0.1, and NVIDIA driver 396.37 on Ubuntu 16.04 OS.

## 3. Vision Recognition Method

The basic rule of the alpine skiing game is to transit all gates. Therefore, DIANA, the skiing robot, must be able to accurately recognize all gates.

### 3.1. Color-Based Recognition

The flags can be accurately recognized using highly visible colors. For the same reason, the skier’s clothes, safety nets, and signs are generally similar in color to the flags. Objects of similar color can be misidentified as flags. In addition, owing to the outdoor conditions, the consistency of the image data cannot be guaranteed [36]. Color-based recognition technology has low outdoor reliability. Therefore, the proven convolutional neural network (CNN), DetectNet, was used [37].

### 3.2. Deep Learning Gate Detection

DetectNet [38], developed by NVIDIA, provides reliable performance when inferring a single class. Therefore, DetectNet is suitable for use by DIANA, wherein only the flag class needs to be determined. In the case of DIGITS, a GUI provided by NVIDIA, the learning results can be intuitively checked. As such, the appropriate learning option value can be quickly determined.

A total of 10,425 images were randomly collected using the Google search engine to train the model. As a condition of a dataset for learning, the image must include a sloping background and one or more articles. This is because images without a substrate during the learning process and the verification stage can yield incorrect learning compensation. In some cases, the initial model incorrectly identified another object as a flag, as shown in Figure 6b.

To eliminate the intermittent misrecognition of safety nets and signs as flags, we collected and trained the field data. As shown in Figure 6a, the final learning model reliably recognized the flag within 8 m. The images were divided into training and validation groups with a 10:1 ratio. Subsequently, 1288 images were collected in the field to increase the recognition rate. As shown in Figure 6c, we obtained a model with a high recognition rate, with a mean average precision (mAP) in excess of 0.9. Loss_box represents the percentage of bounding boxes that was not detected during verification of the learned model. Loss_coverage refers to the degree to which the detected bounding boxes deviated from the actual value. Both items had low error rates of 0.1 or less. DIANA is equipped with a Zed stereo camera and NVIDIA Jetson TX2. Skiing is a fast-moving sport, so image inference must be performed in a short time. However, the NVIDIA Jetson TX2 is not fast enough for inferring image data with a resolution of 1024 × 768 or higher. Therefore, DIANA used 672 × 376, the minimum resolution supported by the ’Zed Stereo Camera’, and achieved a frame rate of 12.

## 4. Motion Generation

### 4.1. Skiing Strategy

During skiing, several turning techniques are performed, including pflug bogens, Stemm turns, and parallel turns. In this study, the ’carved turn,’ with the lowest deceleration and the fastest speed, was selected. This maneuver is a turning method that utilizes the ski side, so, unlike other turns that utilize the ski surface, there is no lateral sliding phenomenon, so the speed is maximized. Instead, because it mainly uses the side of the ski, it is likely to result in falling because of its low stability. Therefore, high control performance is required for driving stability. To perform a ’carved turn,’ we created poses and motion that provided connections between poses. First, we analyzed the poses of a human skier and selected poses, a core of turning motion, and neutral motion, to change the direction of turning. The robot’s poses that were most similar to human poses were selected. The motions were produced using the fifth polynomial trajectory for a natural transition of the continuous poses. Figure 7 shows the motion and role of each turning section for DIANA’s carved turns. Second, we need to decide the path plan, i.e., when to start and finish turning to drive through the desired route. In this study, the path planner receives the distance between DIANA and the gate from the stereo camera and LiDAR sensor, and, based on the distance, the path planner determines the motion of DIANA. For example, when DIANA poses in neutral motion, as shown in Figure 7a, if the gate is located on the right-hand side based on the driving direction, the right turn starts. After this, the LiDAR sensor senses that the gate passes behind DIANA during turning and the distance exceeds the specific distance; the turn motion is finished and moved to neutral motion, as shown in Figure 7c. The distance is specified through repeated experiments. In the simulation, the distance between DIANA and the gate is not measured separately, but the motion of DIANA is determined by receiving the position data of DIANA through dynamic calculations.

### 4.2. Carved Turn

Given that the ski is elastic and has a side-cut structure (a wide front and rear and a narrow center), pressure is applied to its middle section in an edged state (with the ski-board against the surface), resulting in bending of the ski, and causing curvature. Transitioning along this curvature naturally leads to turning [39].

Moreover, owing to the structure of the large side-cuts of the carving ski, the importance of the center of gravity in performing carved turns is high. It is important to use a posture that applies pressure to the middle of the ski without the center of gravity moving forward–backward or left–right during the turn [40]. Therefore, we need to control the ZMP on the x-axis (forward–backward) and y-axis (left–right). However, the ZMP on y-axis is considered more dominant on the stability. Thus, we only controlled the y-axis of the ZMP to reduce the number of variables that needed to be controlled.

The pressure transmitted to the ski plate during the carved turn is generated from the snow surface, produced by an edge operation (tilting the ski plate relative to the ground) via a centripetal force. To maintain this force, as shown in Figure 7b, the inclination in which the center of the body is located inside the turn compared to the lower body and angulation in which the lateral movements of creating angles at the waist is important [41,42]. It is ideal for the lateral ZMP (zero moment point) to remain in the center of both feet during inclination and angulation. However, in actual skiing, the edge, which provides the main centripetal force to the snow surface, is strongly formed on the outside foot and most of the snow surface is uneven. Therefore, starting with the outside foot is recommended: inner foot = 10:0, converge and maintain at 7:3 [43]. In the case of the robot, the weight of the upper body was much lower than the weight ratio of an individual upper body, so the center of gravity was not sufficiently outside the foot (foot in the progress of turning). Therefore, in addition to the movement of tilting the upper body outward, the movement of pushing the outer foot forward was combined so that the center of gravity was loaded on the outer ski. Human skiers have limited ankle movement due to ski boots. Unlike human skiers, DIANA can move its ankles. Thus, DIANA can flexibly adjust the ski edges for the carved turn as needed. DIANA has the advantage of being able to use a carved turn without moving as much as a human.

After performing the carved turn, the bent ski returned to its original state and generated a repulsive force, which required a movement to connect the repulsive force to the next turn. In the case of DIANA, neutral motion was inserted between each carved turn, as shown in Figure 7a,c, to absorb the rebound caused by the carved turn. Neutral motion is a movement of instantly returning angulation and pivoting (an action of internally rotating the outside foot), gradually performed during the carved turn process, to the initial state before turning it [44,45]. In the case of DIANA, neutral motion occurred during the absorption of the reaction force by slightly bending both legs.

### 4.3. Dynamics of a Carved Turn

To stably drive a robot, the zero moment point (ZMP) must be inside a support polygon containing both skis. If the ZMP deviates from the outside of the support polygon, one of the skis is likely to lose contact with the ground, resulting in a loss of stability. According to the general definition of ZMP [46], the components of the ZMP are represented by Equation (Equation 1).
(1)$$\begin{array}{cc} x_{ZMP} & =\frac{\sum m_{i}{\ddot{z}}_{i}x_{i}-\sum m_{i}{\ddot{x}}_{i}z_{i}-\sum {\left(T_{y}\right)}_{i}}{\sum m_{i}{\ddot{z}}_{i}},\;{\ddot{x}}_{i}=g_{x},\;{\ddot{z}}_{i}=g_{z}\\ y_{ZMP} & =\frac{\sum m_{i}{\ddot{z}}_{i}y_{i}-\sum m_{i}{\ddot{y}}_{i}z_{i}-\sum {\left(T_{x}\right)}_{i}}{\sum m_{i}{\ddot{z}}_{i}},\;{\ddot{y}}_{i}=g_{y}\pm \frac{\upsilon^{2}}{R_{i}},\;{\ddot{z}}_{i}=g_{z} \end{array}$$

To calculate the ZMP, the location, speed, and acceleration of the skiing robot on the driving route must be calculated. Unlike linear driving, specific dynamic modeling is required because the force applied to the robot varies depending on the turning angle.

Figure 8 shows a vertical view with respect to the ski slope. Suppose that the force acting on the skiing robot is the force of gravity and friction between the ski and the surface. If the coefficient of friction is set as μ and the inclination angle of the slope is set as α, the acceleration acting on the robot is given by Equation (Equation 2).
(2)$$\begin{array}{cc} \ddot{x} & =g\sin\alpha \cos\psi -\mu g\cos\alpha \\ \ddot{y} & =\pm r\omega^{2}\\ \ddot{z} & =g\cos\alpha \end{array}$$

In the case of the angular speed ω, it depends on the rotation angle, ψ, so an expression for the relationship between ω and ψ is needed. Given that the tangential acceleration for rotation is $\ddot{x}$, when the speed at which rotation occurs is represented as $\upsilon_{0}$, ω can be expressed as in Equation (Equation 3).
(3)$$\dot{\psi }=\omega =\int_{0}^{t}\frac{\ddot{x}}{r}dt=\frac{\upsilon_{0}}{r}+\frac{1}{r}\int_{0}^{t}g\sin\alpha \cos\psi -\mu g\cos\alpha dt$$

The radius of curvature *r* is determined for the case in which the ski is bent by an amount given by the edge angle of the ski θ during the carved turn, which is represented in Figure 9. The relationship between the radius of curvature and the edge of skiing can be expressed as in Equation (Equation 4), referred to as the ’The New Skiing Mechanics: Including the Technology of Short Radius Carved Turn Skiing and the Claw’ by Howe [47].
(4)$$r=\frac{\frac{L^{2}\cos\theta }{4}+\frac{h^{2}}{\cos\theta }}{2h}$$

## 5. Controls for Skiing

DIANA’s system was implemented using the robot operating system (ROS) packages for each function. Every sensor publishes the data as an ROS message and collects and uses the data in packages that require data.

DIANA’s control method can be divided into high-level and low-level control. High-level control determines which posture is adopted depending on the situation, based on the data collected by sensors as DIANA descends a slope. In low-level control, DIANA calculates the target position of the motor located in each joint every 8 ms to realize a specific posture and transmits the command. The aforementioned two control methods were executed on separate CPUs to minimize any delays associated with the temporary unavailability of CPU resources. This delay may cause unintentional movement during motion, which can result in an accident.

Figure 10 shows a diagram of the software structure of DIANA.

The high-level control determines the carved turn timing by referring to the vision inference data. A color image with a size of 672 × 376 is acquired from the left camera of the Zed stereo camera. It is transmitted to DetectNet, which infers flags present in the image using a pre-trained neural network model.

DetectNet transmits the detected objects to the DIANA Commander, which obtains object position data from the Zed stereo camera and LiDAR. The positions of the flags detected via inference are estimated using the bounding box and the depth map. To increase the reliability, the value is compensated using the distance information acquired from the LiDAR sensor. Depending on the position of the flags acquired via the aforementioned process, the DIANA Commander transmits motion commands to the DIANA Motion Module.

Figure 11 is a flowchart that shows the process utilized by the DIANA commander in determining motion based on the sensor data. When the DIANA Commander is initiated, the results of vision recognition are collected. If the flag is not found, vision recognition is performed again. If the flag is found, the gate location is confirmed. If the gate is located sufficiently far away, the position of the gate is located relative to the sides of DIANA. A straight pose is assumed if the gate is located in the middle relative to DIANA. However, a left-turn pose is assumed if the gate is located on the left of DIANA. Otherwise, a right-turn pose is assumed. If the gate is sufficiently close, the robot performs a scan using LiDAR. If the gate is detected, a straight pose is initiated. A left-turn pose is initiated if the gate is on the right side of the slope; otherwise, a right-turn pose is assumed. After a pose is adopted, the vision recognition process is reinitiated. Low-level control directly sends a command to the motor to assume a posture according to the high-level control command. The DIANA Motion Module transmits the target position to the motor controller so that the robot can assume a specific posture according to the command transmitted from the DIANA Commander. In this case, the IMU sensor and the F/T sensors measure the ZMP that moves out of the support polygon owing to an irregular external force and compensates for it by moving the center of mass.

## 6. Simulation

### 6.1. Simulation Method

Based on dynamic modeling, a simulation of skiing robots was conducted using MATLAB R2017a(MathWorks). The simulation proceeded in the same order as shown in Figure 12.

The driving path of the skiing robot was set as an ideal path, as shown in figure, based on the magnitude of the slope and the location of the gate. Assuming that there are no external forces other than friction between the ski and snow, the simulation proceeds as follows.

1.The motion of DIANA is determined by the path planner according to the path set by the skiing robot, and the rotation angle ψ and angular velocity $\dot{\psi }$ of DIANA are calculated.2.In the DIANA Motion Module, $\theta_{joint}$ is determined by calculating the IK of DIANA based on the corresponding motion information.3.In the dynamics calculation module, the ZMP is calculated using ψ, $\dot{\psi }$, and $\theta_{joint}$ obtained from the path planner and DIANA Motion Module.4.The dynamics calculation module feeds back the radius of curvature of the ski $r_{ski}$, DIANA’s position *p*, velocity $\dot{p}$, and acceleration $\ddot{p}$ to the planner.

The main issue in skiing is maintaining stability in the left and right directions to prevent falling. The stability in the front and rear directions is high because of the length of the ski and is not significantly affected by changes in the ski motion. However, even though the left and right directions are significantly affected by external forces such as the centripetal force and gravity during rotational driving, the support side is very narrow, and thus is not stable. Therefore, it is necessary to focus on the stability in the left and right directions. In general, humanoid robots use ZMP for stability analysis [48,49]. The skiing robot may also calculate stability during driving by determining the presence or absence of the ZMP inside the support polygon made by the ski owing to the ZMP [50,51].

### 6.2. Simulation Result

The simulation was conducted in an environment similar to the actual experimental environment of the skiing robot. The slope angle was set to $15^{\circ }$, and a ski with a minimum rotation radius of 8.7 m was used. Figure 13 shows an example (right) of the robot’s motion and the calculated ZMP at the location indicated on the skiing robot’s path (left) in the simulation. Figure 14 shows the change over time for $ZMP_{y}$. The simulation in this study was performed using ZMP based on the center point between both feet of DIANA during skiing. In this case, if the ZMP in the lateral direction is inside the support polygon formed by both skis of DIANA, it can be established that the robot is in a stable state, so the positions of $ZMP_{y}$ and the ski plates, which are elements in the lateral direction of ZMP, are as shown in Figure 14.

In the experiment, DIANA was pushed directly to the starting point. Therefore, assuming that the initial velocity is one factor that affects the result, the simulation was conducted by changing the initial velocity. If the initial velocity is low, as in Figure 14 (green), it is evident that $ZMP_{y}$ is stable between the two skis. However, in this case, the turn starts relatively late compared to the other cases, and more time is required for completion. However, if the initial velocity is high, such as in Figure 14 (blue), the turn starts relatively early compared to the other cases, which implies that less time is required for completion. If a third turn is performed, $ZMP_{y}$ is out of the ski plate. Thus, it can be predicted that driving will be relatively unstable compared to the other cases. In the simulation results, the appropriate initial velocity value was approximately 4.8 m/s. In the actual test, several variables can have an effect, so identical results are not necessarily expected when tested under the same conditions. However, initiating DIANA at an appropriate speed has a significant influence on stability.

## 7. Field Test

DIANA was used to conduct a field test at Phoenix Park, located in Pyeongchang, Korea, on 12 February 2018, to evaluate its performance. Phoenix Park is characterized by a low temperature and a large amount of snow on a site of 6.6 × 10^6^ m^2^. It is renowned for its excellent snow removal ability and snow quality among ski resorts in Korea.

### 7.1. Environment

On the day of the experiment, the average wind speed was 5 m/s, the minimum temperature was −16 °C, and the maximum temperature was −7 °C. Considering the performance of DIANA, a total of five gates were installed on an 80 m long slope based on the rules of the original alpine competition. As shown in Figure 15, the experiment was conducted on an actual ski slope with an average inclination of 15°.

### 7.2. Test Result

Figure 16 shows a chronological list of the scenes wherein DIANA recognized the flags as it traveled down the slope. DIANA correctly recognized the flags while moving down the slope and did not misidentify any objects.

Figure 17 shows the skiing motions of DIANA on the slopes, arranged according to time. It was confirmed that DIANA was able to drive on the slope by transiting through the gates while performing a carved turn. Figure 18 shows the y-axis value of ZMP calculated based on the force data measured using the F/T sensors installed on both feet, while DIANA was in motion, and the y-axis of ZMP calculated using the simulation results. This is a comparative graph. It is evident in Figure 17 that DIANA is performing neutral motion in approximately 1.8 s to 2.6 s, carved turn motion in approximately 2.6 s to 4.8 s, and neutral motion again in approximately 4.8 s to 5.6 s. The aforementioned motions can be represented by areas Figure 18a, Figure 18b, and Figure 18c and are consistent with Figure 7a, Figure 7b, and Figure 7c.

In section (a), it is evident that a weight shift occurs during ’neutral motion’. In the (b) section, a full ’carved turn’ occurs, and in the beginning, it is evident that $ZMP_{y}$ moves toward the ski plate corresponding to the outer foot as the outer foot pressure increases. In the second half, $ZMP_{y}$ returns to the baseline. A weight shift occurs during neutral motion in section (c), and $ZMP_{y}$ moves to the ski plate corresponding to the outer foot as a ’carved turn’ is performed again after section (c).

However, the overall $ZMP_{y}$ enters the support polygon that constitutes the ski plate of both feet, and a continuous ’carved turn’ is successfully performed without falling, even in actual skiing.

## 8. Conclusions

In this paper, we present the research process and results for a humanoid robot with a height of 1.23 m as it participated in a human giant slalom match on an actual slope. The flags were recognized based on the stereocamera’s image information using ’CNN&DetectNet’, distances to flags were measured using LiDAR sensors, and the path for the robot to ski was created based on the flags’ locations. For the robot to follow a given path, appropriate rotational and switching motions were created by imitating the posture of humans skiing, and a stable carved turn was used by compensating for motions in real time based on information from F/T sensors and an IMU sensor so that the robot did not fall despite various environmental changes. A numerical simulation tool was developed using MATLAB R2017a to evaluate the validity and stability of the carved turn motion prior to the actual slope experiment and the development of the robot.

Finally, the motion of DIANA in simulation analysis was verified via experimentation on the actual slope.

We created a learning model by utilizing a convolutional neural network for flag recognition. The artificial intelligence model presented in this study stably recognized flags even in outdoor environments under various weather conditions.

In addition, unlike previous studies, a humanoid robot with a height of 1.23 m and 23-DOF was used to autonomously complete actual giant slalom skiing tasks with carved turns, similar to human skiers on actual slopes.

The performance of the robot on various human skiing tasks was evaluated as successful if tasks that are difficult for humans to perform were conducted in harsh environments. As such, the results of this study advance the technology of humanoid robots by closing the gap between the abilities of humanoid robots and their human counterparts. If the humanoid skiing robot continues to be developed, it could be used in the future as a means to save lives if people are isolated or trapped in snowy mountains.

However, this research is limited to implementing the same level of turning as a human skier, because DIANA does not control the ZMP in the x-axis (forward–backward) and does not consider damping disturbance caused by uneven snow surface. Therefore, in outdoor experiments, DIANA falls sometimes depending on the surface conditions. In the future, if the ZMP control of the x-axis can also be performed to identify and correct the effects on stability, it is expected that the humanoid robot will be able to perform a more stable carved turn by overcoming the unevenness of the snow surface. Based on this study, we intend to continue the development of skiing robots with higher stability and faster speed by investigating slopes with higher difficulty. Just as Robocup aims to outperform humans at the highest level in soccer, we intend to continue to pursue research on skiing robots with the aim of breaking human skiing records at the Winter Olympics.

## Figures and Tables

**Figure 1 sensors-22-00816-f001:**
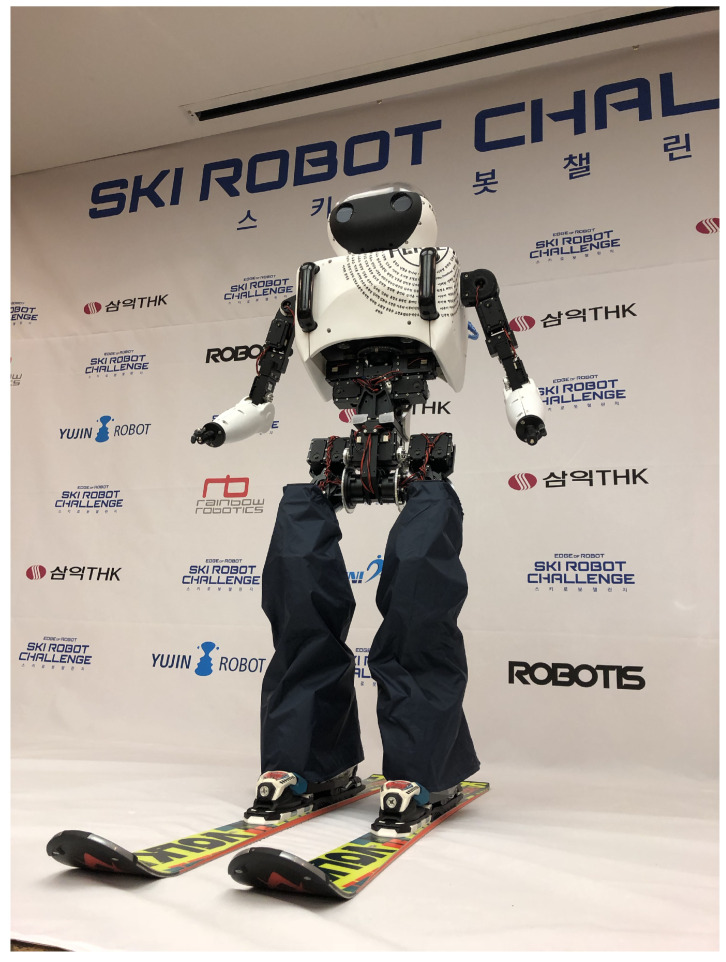
Skiing robot DIANA.

**Figure 2 sensors-22-00816-f002:**
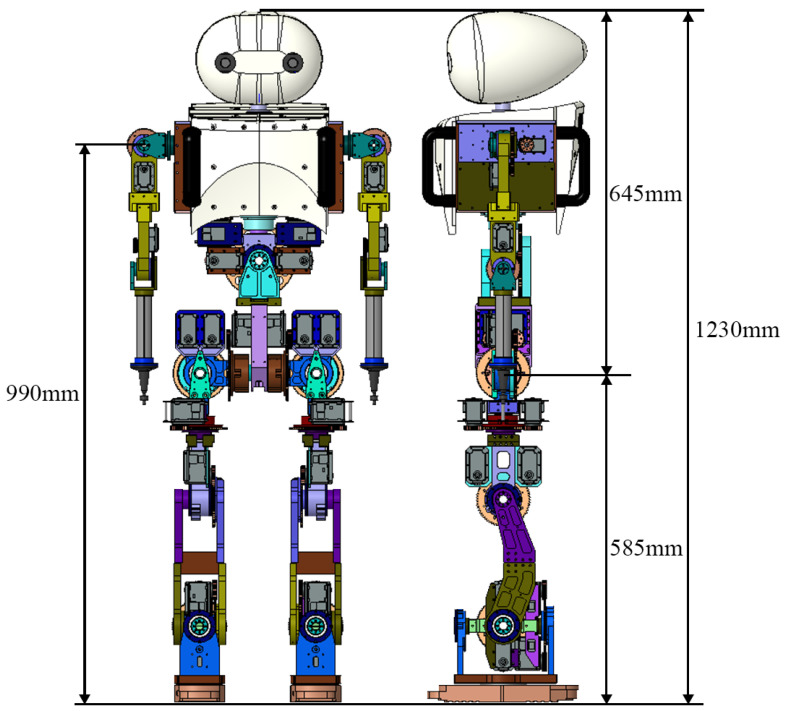
Three-dimensional design of skiing robot DIANA.

**Figure 3 sensors-22-00816-f003:**
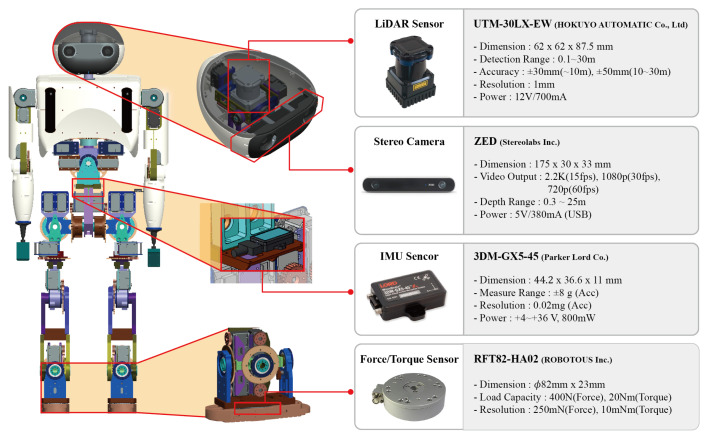
Design of DIANA’s sensors.

**Figure 4 sensors-22-00816-f004:**
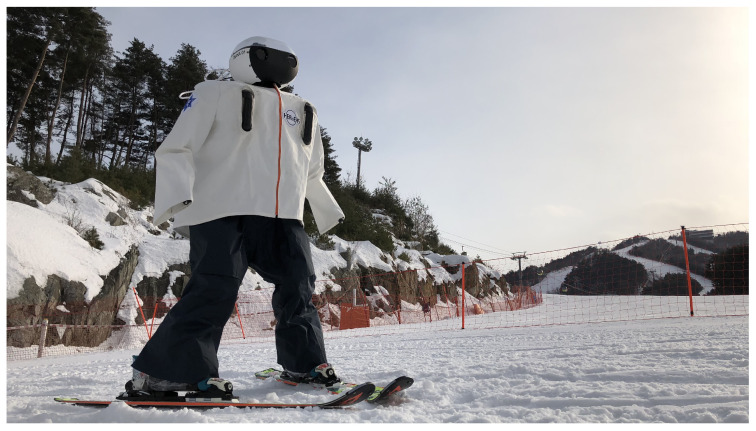
DIANA’s waterproof design.

**Figure 5 sensors-22-00816-f005:**
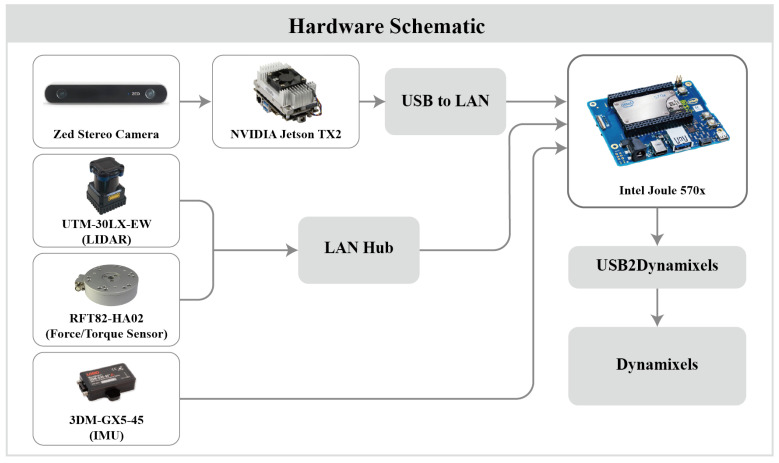
Electrical component diagram of skiing robot DIANA.

**Figure 6 sensors-22-00816-f006:**
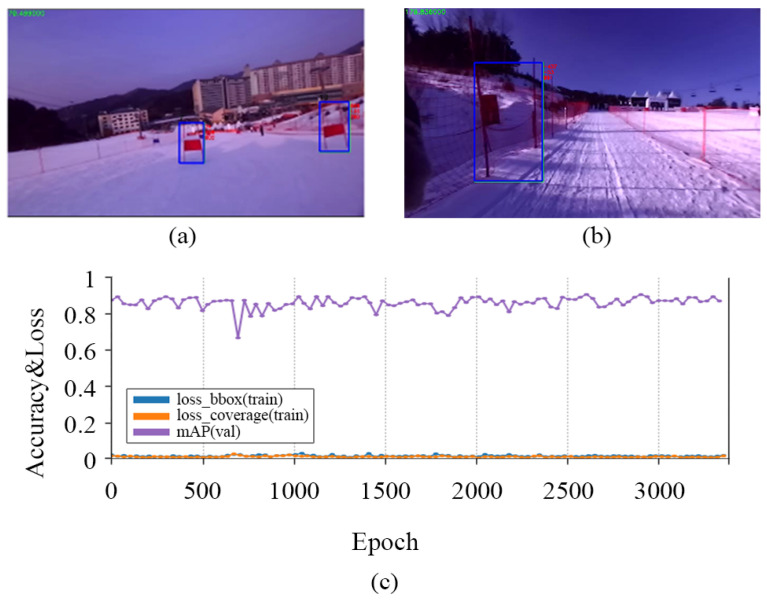
The correct result of recognition (**a**), the incorrect result of recognition (**b**) and the result graph of training (**c**).

**Figure 7 sensors-22-00816-f007:**
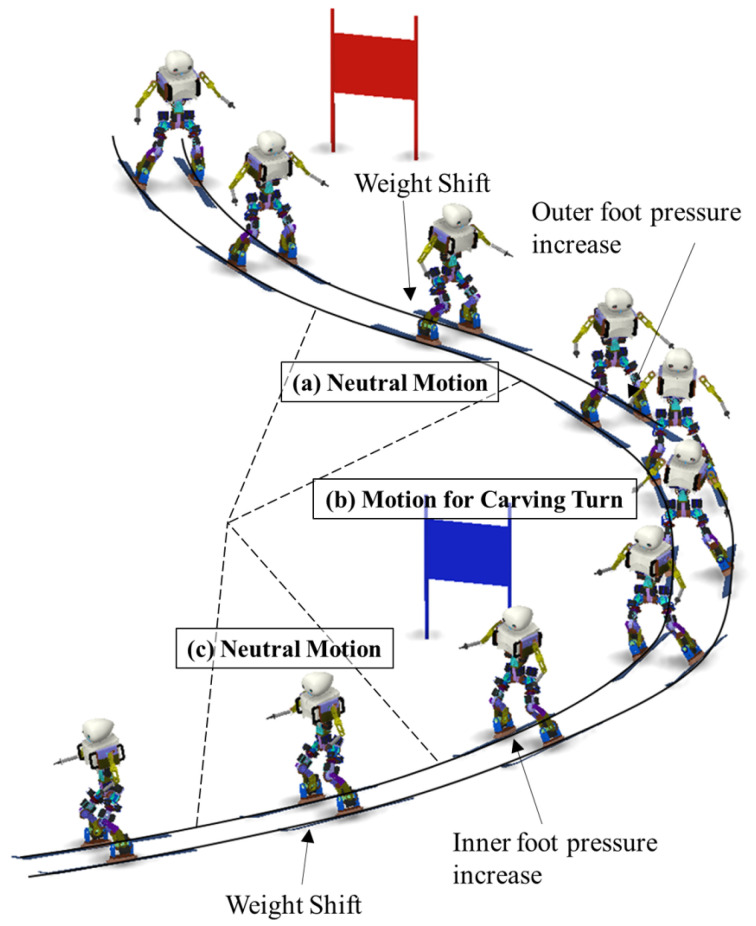
Motion and role of each turning section for DIANA’s carved turns.

**Figure 8 sensors-22-00816-f008:**
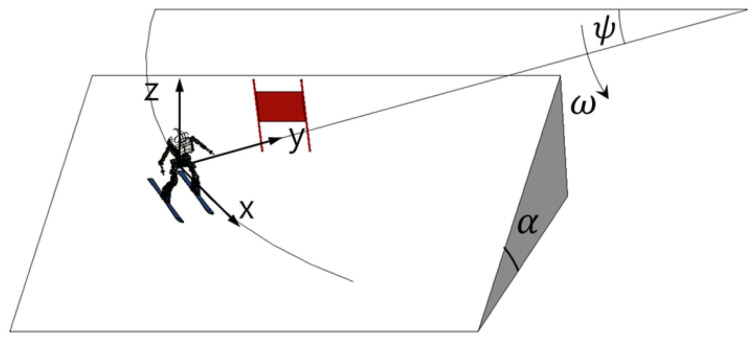
Top view of ski slope.

**Figure 9 sensors-22-00816-f009:**
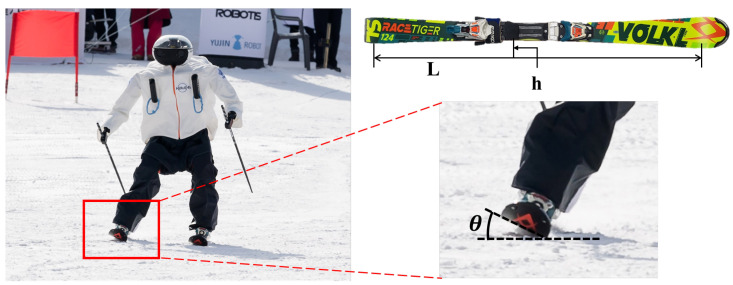
Curvature radius.

**Figure 10 sensors-22-00816-f010:**
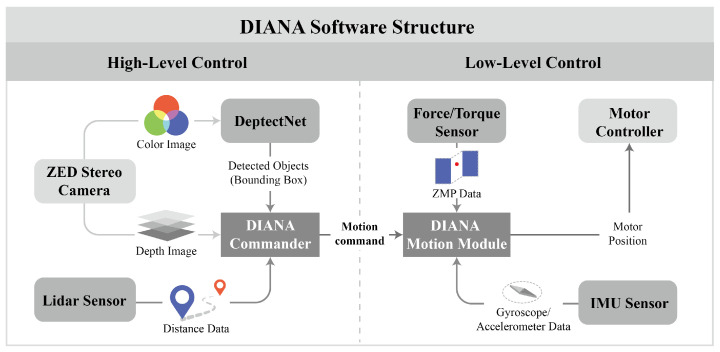
Diagram of DIANA’s software structure.

**Figure 11 sensors-22-00816-f011:**
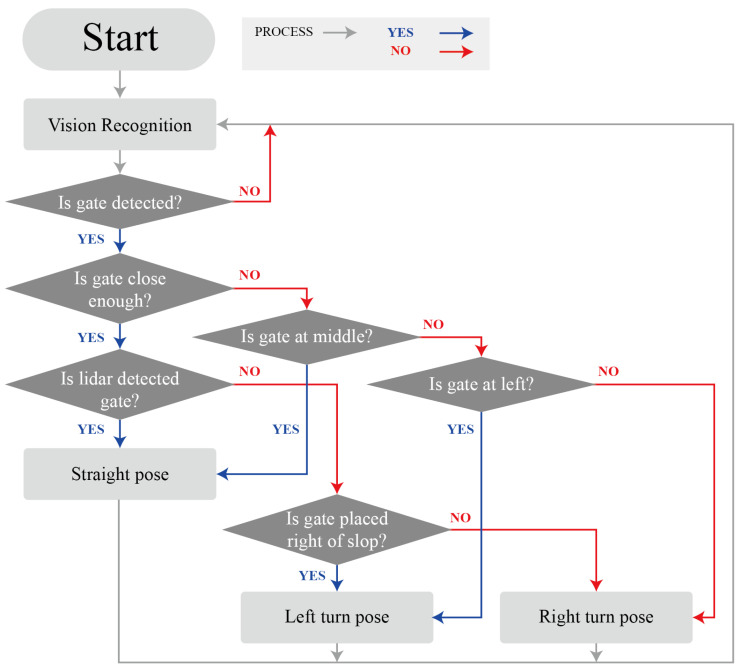
Flow chart for DIANA commander.

**Figure 12 sensors-22-00816-f012:**
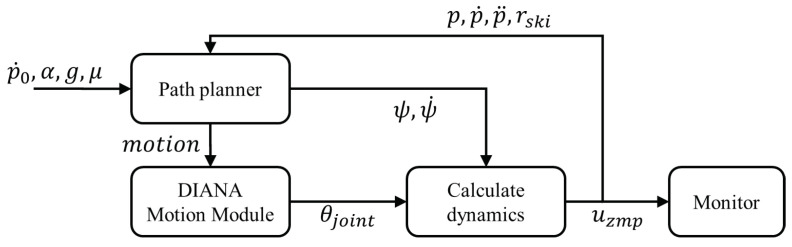
Simulation process.

**Figure 13 sensors-22-00816-f013:**
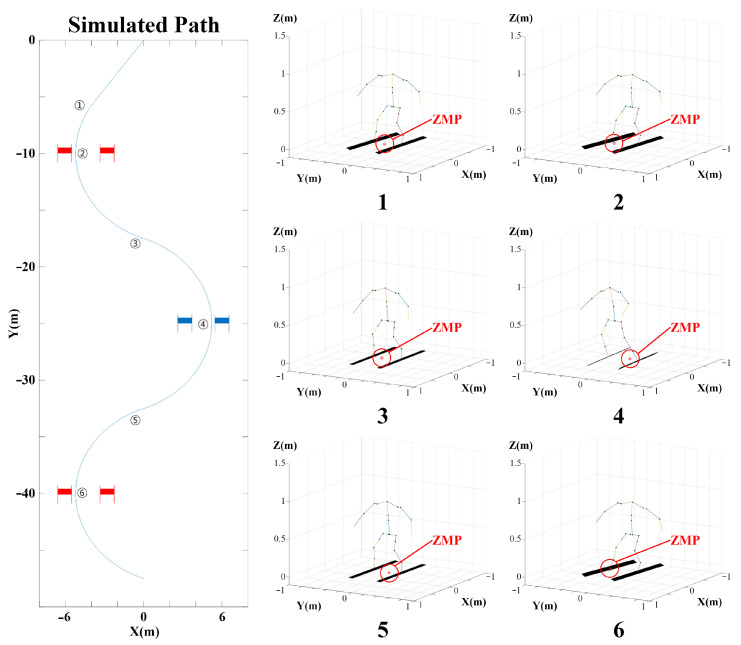
DIANA simulation path (**left**) and DIANA’s motion with ZMP (**right**).

**Figure 14 sensors-22-00816-f014:**
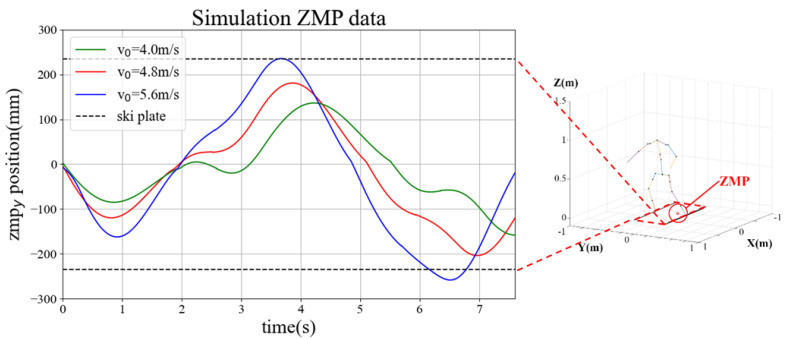
$ZMP_{y}$ variants according to time.

**Figure 15 sensors-22-00816-f015:**
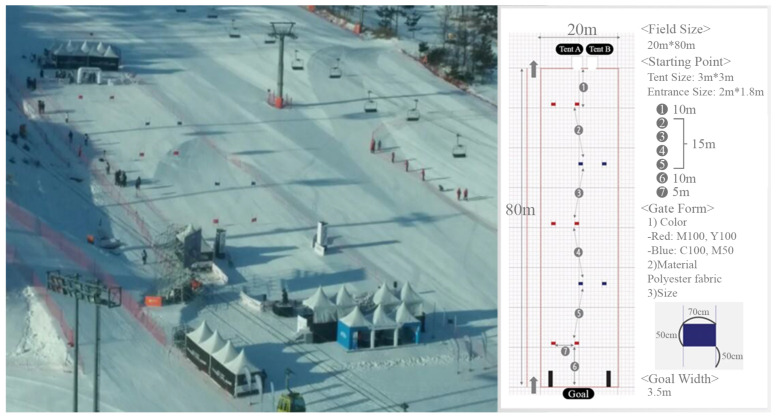
A view of the slope (**left**) and a plan view (**right**).

**Figure 16 sensors-22-00816-f016:**
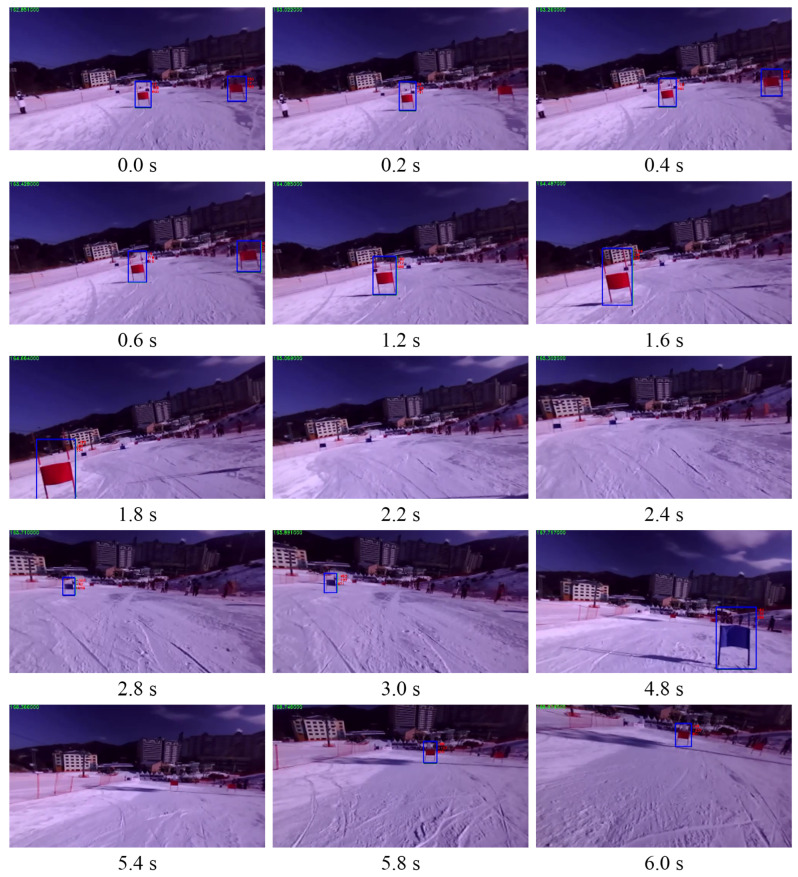
Recognition result of skiing DIANA.

**Figure 17 sensors-22-00816-f017:**
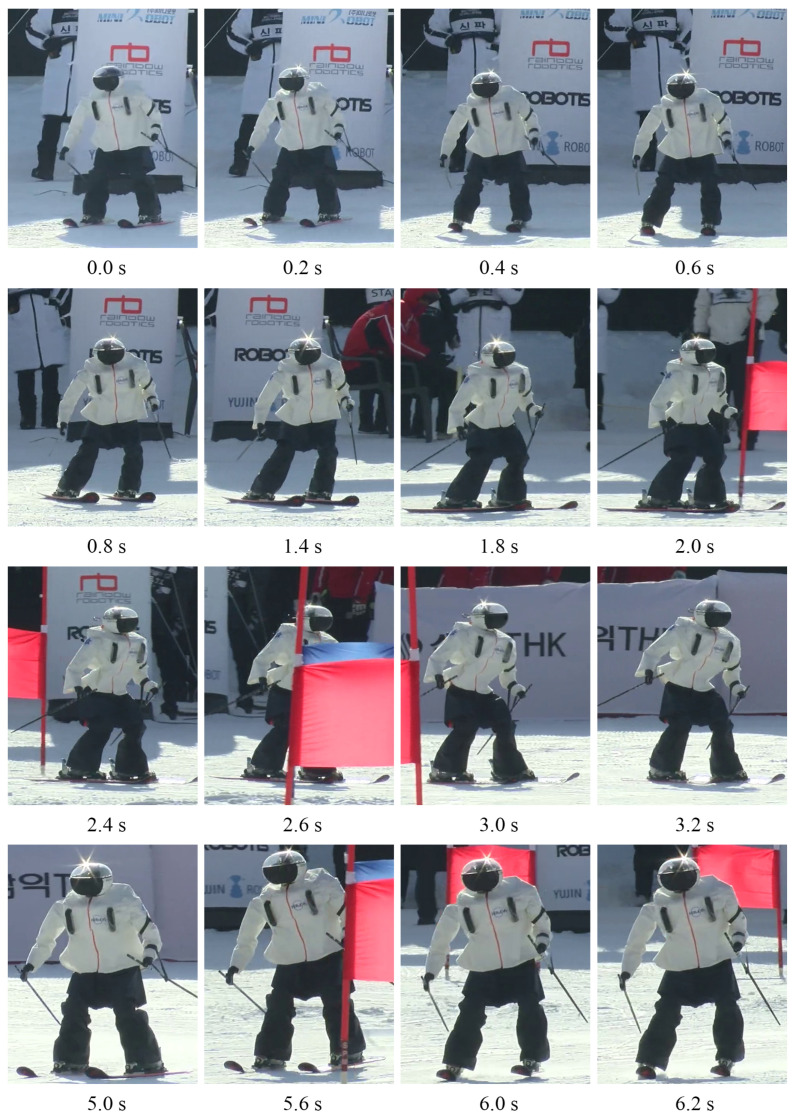
The scene wherein DIANA travelled down the slope.

**Figure 18 sensors-22-00816-f018:**
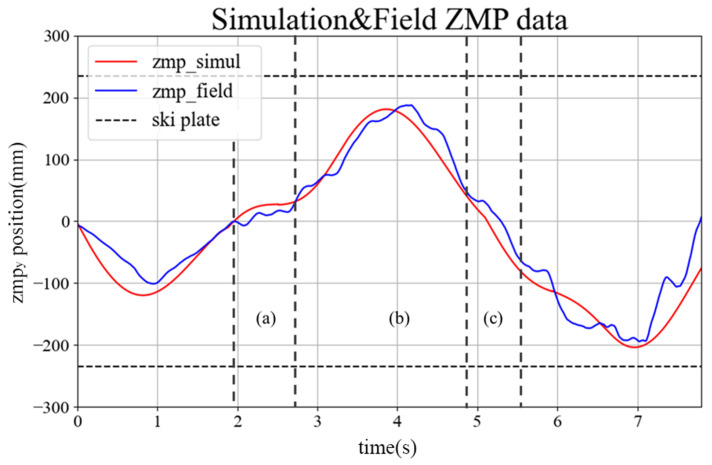
Comparison of $ZMP_{y}$ based on simulation and field test results.

**Table 1 sensors-22-00816-t001:** DIANA’s degrees of freedom (DOFs).

	Degrees of Freedom (DOFs)		
	**Left**		**Right**
Head		3	
Arm	3		3
Waist		2	
Hip	3		3
Knee	1		1
Ankle	2		2
Total		23	

## Data Availability

Not applicable.

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
