# Peer review of "Carved Turn Control with Gate Vision Recognition of a Humanoid Robot for Giant Slalom Skiing on Ski Slopes"

_sensors, 2022, doi:10.3390/s22030816_

Round 1

Reviewer 1 Report

The paper was a fun paper to read.  This reviewer happens to be an expert skier and former racer.

The paper is well written.  The methods are well-described.  There are some areas where the paper could be improved.

Clearly state the innovations of the work.  The vision and object identification use fairly standard methods.  The control of the robot is a quasi-static ZMP method.  Explain the relevance of the work to more general robotics work, essentially: who cares about a skiing robot?

In section 6.1, the process is presented in 4 steps.  Step 3 says that a Pathplanner defines the parameters.  A description of the path planner is needed.  The path used by the robot is not very optimal for racing.  In human alpine racing, the coach spends most of the training on the path used by the racer.

The control of the ski edge seems to be determined by the ankle joint (looking at figure 17, 6.0 and 6.2 sec).  This is very different from human skiers wearing stiff boots.  There also appears to be some ski bouncing on the snow.  Does this occur?  Human skiers use their legs (knees) to control the vibration and bouncing of the ski on the snow.  Human skiers also do not just shift weight to turn but weight and unweight the skis for the turn.  The robot does not seem to have any compliance in the legs nor employ weight/unweight.  Other skiing robots have used a simple weight shift for turning.  Explain the innovation and difference for this robot.

It is stated that it is required to have the pressure at the center of the ski (line 221).  Human skiers move the pressure forward and backward on the ski depending on the snow conditions, slope, and dynamics of the turns.

It would be good to add any future work that is planned, such as using the action of the legs and more dynamic ski control that humans use.  So far, the robot rates as a beginner skier.  What would it take to become an expert?

Author Response

Point 1: Clearly state the innovations of the work. The vision and object identification use fairly standard methods. The control of the robot is a quasi-static ZMP method. Explain the relevance of the work to more general robotics work, essentially: who cares about a skiing robot?

Response 1: I have added the innovation of the work at Introduction (line 107-111) and the importance of developing skiing robot at Conclusion (line 436, 437). 

Point 2: In section 6.1, the process is presented in 4 steps.  Step 3 says that a Pathplanner defines the parameters.  A description of the path planner is needed.  The path used by the robot is not very optimal for racing.  In human alpine racing, the coach spends most of the training on the path used by the racer.

Response 2: I have described about the path planner at 4.1. Skiing Strategy section (line 216-227). 

Point 3: The control of the ski edge seems to be determined by the ankle joint (looking at figure 17, 6.0 and 6.2 sec).  This is very different from human skiers wearing stiff boots.  There also appears to be some ski bouncing on the snow.  Does this occur?  Human skiers use their legs (knees) to control the vibration and bouncing of the ski on the snow.  Human skiers also do not just shift weight to turn but weight and unweight the skis for the turn.  The robot does not seem to have any compliance in the legs nor employ weight/unweight.  Other skiing robots have used a simple weight shift for turning.  Explain the innovation and difference for this robot.

Response 3: I have added about difference of ankle between human skier and DIANA at 4.2. Carved Turn section (line 254-257). Also added advantage that the ankles of DIANA can get. 

Point 4: It is stated that it is required to have the pressure at the center of the ski (line 221).  Human skiers move the pressure forward and backward on the ski depending on the snow conditions, slope, and dynamics of the turns.

Response 4: I have explained why I didn’t control the ZMP on x-axis at 4.2. Carved Turn section (line 236-239). 

Point 5: It would be good to add any future work that is planned, such as using the action of the legs and more dynamic ski control that humans use.  So far, the robot rates as a beginner skier.  What would it take to become an expert?

Response 5: I have added future work about to improving DIANA to expert skier level at Conclusion section (line 438-444). 

Reviewer 2 Report

The paper is clear and much interesting for readers, contain significant results, back results by experiments. But I guess the paper should be major revised.

1. The Abstract is unallowable. Firstly it contain too many details such as enumeration of problems which should be solved under the humanoid robot skiing. Secondly, please allocate actuality of the researching problem. The Abstract should be short and substantial : the solving problem formulation, the method of research and brief results.

2. Please place the references in the first two paragraph of the Introduction. Back propositions by references.

In general I guess the paper should be carefully readen once again. It contain a lot of small but significant misprints.

1) Please replace the record such as "[1][2]" by "[1,2]" in many places.

2) Please correct misprint at the 80 line of the 2 page: "win.[21][22]" -> "win [21-22]". Such misprint (dot before reference) appears at the many places (see for example, line 170 and 172 at the 6th page).

3) Correct the title of Figure 5 "Haedware Schematic"

4) Not all sources in the bibliography contain pages ("pp." or "p."). Alongside, not all years are bolded. Please read the bibliography through carefully once more.

5) Please correct the misprint at the 262-263 line at the 9th page.

6) At the line 378 of the 15th page the word "Figure" appears twice. Correct it.

Please read the paper once again and correct small misprints.

Author Response

Point 1: The Abstract is unallowable. Firstly it contain too many details such as enumeration of problems which should be solved under the humanoid robot skiing. Secondly, please allocate actuality of the researching problem. The Abstract should be short and substantial : the solving problem formulation, the method of research and brief results.

Response 1: I have rewritten the part of Abstract (line 7-17). I got rid of the enumeration of problems part, and added about goal of the paper with how to solving problems. 

Point 2: Please place the references in the first two paragraph of the Introduction. Back propositions by references.

Response 2: I added two more references at first two paragraph of the Introduction (line 22, 34). 

Point 3: In general I guess the paper should be carefully readen once again. It contain a lot of small but significant misprints. 
1) Please replace the record such as "[1][2]" by "[1,2]" in many places.
2) Please correct misprint at the 80 line of the 2 page: "win.[21][22]" -> "win [21-22]". Such misprint (dot before reference) appears at the many places (see for example, line 170 and 172 at the 6th page).
3) Correct the title of Figure 5 "Haedware Schematic"
4) Not all sources in the bibliography contain pages ("pp." or "p."). Alongside, not all years are bolded. Please read the bibliography through carefully once more.
5) Please correct the misprint at the 262-263 line at the 9th page.
6) At the line 378 of the 15th page the word "Figure" appears twice. Correct it.
Please read the paper once again and correct small misprints.

Response 3: I corrected the misprints that you found (number 1-6). Also I read the whole paper clearly and corrected every small misprints. 
I have corrected the missing page information in some references.
I tried changing the year to bold in References as you said, but the MDPI Latex format is already defined. So changing the year to bold causes a lot of errors. Only the year of the article reference is shown in bold.

Reviewer 3 Report

A very well written paper, I am particularly impressed by the inclusion of data such as environmental factors which are significant variables for the reproducibility of this research. 

Figures are clear with appropriate notations.

I had trouble reading figure 15 due to the size of the text in the figure, I would suggest re-scaling this image to make the text easier to read.

In the introduction you talk about the application for your system in everyday scenarios rather than just skiing, it would be useful to re-state this in the conclusion and maybe give a scenario were your system could be implemented to build on the current state-of-the-art.

Overall this is a good quality paper, very interesting, innovative and relevant. 

Author Response

Point 1: I had trouble reading figure 15 due to the size of the text in the figure, I would suggest re-scaling this image to make the text easier to read.

Response 1: I have increased the text size in Figure 15 and also resized the image for better readability. 

Point 2: In the introduction you talk about the application for your system in everyday scenarios rather than just skiing, it would be useful to re-state this in the conclusion and maybe give a scenario were your system could be implemented to build on the current state-of-the-art.

Response 2: I have added the content in the Conclusion section (line 438-444) that to further develop the current DIANA to compete with human skiers. 

Round 2

Reviewer 2 Report

All my suggestions was taken into account. So I guess the paper may be published.